# Identification of a Novel Gene *MtbZIP60* as a Negative Regulator of Leaf Senescence through Transcriptome Analysis in *Medicago truncatula*

**DOI:** 10.3390/ijms251910410

**Published:** 2024-09-27

**Authors:** Jiayu Xing, Jialan Wang, Jianuo Cao, Ke Li, Xiao Meng, Jiangqi Wen, Kirankumar S. Mysore, Geng Wang, Chunjiang Zhou, Pengcheng Yin

**Affiliations:** 1Ministry of Education Key Laboratory of Molecular and Cellular Biology, Hebei Research Center of the Basic Discipline of Cell Biology, Hebei Collaboration Innovation Center for Cell Signaling and Environmental Adaptation, Hebei Key Laboratory of Molecular and Cellular Biology, College of Life Sciences, Hebei Normal University, Shijiazhuang 050024, China; 15097315361@163.com (J.X.); wangjialan0312@163.com (J.W.); c13082348698@163.com (J.C.); like@hebtu.edu.cn (K.L.); mengxiao2012@163.com (X.M.); gengwang@hebtu.edu.cn (G.W.); 2Department of Plant and Soil Sciences, Institute for Agricultural Biosciences, Oklahoma State University, Ardmore, OK 73401, USA; jiangqi.wen@okstate.edu; 3Department of Biochemistry and Molecular Biology, Institute for Agricultural Biosciences, Oklahoma State University, Ardmore, OK 73401, USA; kmysore@okstate.edu

**Keywords:** leaf senescence, *Medicago truncatula*, transcriptome analysis, bZIP, WRKY

## Abstract

Leaves are the primary harvest portion in forage crops such as alfalfa (*Medicago sativa*). Delaying leaf senescence is an effective strategy to improve forage biomass production and quality. In this study, we employed transcriptome sequencing to analyze the transcriptional changes and identify key *senescence-associated genes* under age-dependent leaf senescence in *Medicago truncatula*, a legume forage model plant. Through comparing the obtained expression data at different time points, we obtained 1057 differentially expressed genes, with 108 consistently up-regulated genes across leaf growth and senescence. Gene Ontology and Kyoto Encyclopedia of Genes and Genomes pathway enrichment analyses showed that the 108 *SAGs* mainly related to protein processing, nitrogen metabolism, amino acid metabolism, RNA degradation and plant hormone signal transduction. Among the 108 *SAGs*, seven transcription factors were identified in which a novel bZIP transcription factor MtbZIP60 was proved to inhibit leaf senescence. MtbZIP60 encodes a nuclear-localized protein and possesses transactivation activity. Further study demonstrated MtbZIP60 could associate with MtWRKY40, both of which exhibited an up-regulated expression pattern during leaf senescence, indicating their crucial roles in the regulation of leaf senescence. Our findings help elucidate the molecular mechanisms of leaf senescence in *M. truncatula* and provide candidates for the genetic improvement of forage crops, with a focus on regulating leaf senescence.

## 1. Introduction

Leaf senescence represents the concluding stage of leaf development and is characterized by a transition from metabolism to catabolism. This stage involves the breakdown of organelles like chloroplasts, along with the degradation of biological macromolecule such as nucleic acids, proteins and lipids. The released organic and inorganic nutrients are then allocated to young leaves, developing flowers, seeds or other storage organs within the same plant, which is critical for its adaptability and reproductivity [1,2].

Leaf senescence is a crucial issue in both development biology and agronomy. In crop production, precocious leaf senescence can impact photosynthesis and impede grain filling, resulting in significant yield loss. Conversely, delaying leaf senescence may improve final yield in specific scenarios [3,4,5,6]. It is estimated that postponing the senescence of functional leaves for one day can boost the yield of maize, rice, wheat and other crops by a range of 2–10% [7]. When it comes to forage crops, where leaves are primarily harvested, leaf senescence has a more obvious impact on the biomass yield and forage quality [8]. Therefore, revealing the molecular mechanism of leaf senescence can not only improve the comprehension of this fundamental biological process, but may also provide potential strategies for manipulation of leaf senescence, ultimately benefit crop production.

The initiation and progression of leaf senescence are not passive occurrences, but rather tightly controlled by different regulatory factors at multiple levels, including chromatin remodeling, transcription regulation, post-transcriptional regulation, translation regulation and post-translational regulation [1,9,10]. The onset of leaf senescence is basically determined by age. However, as senescence advances, a combination of extrinsic and intrinsic signals are incorporated into age information to regulate the senescence progression. These factors include nutritional and water status, hormone signals, light and temperature conditions and various biotic stresses. As a result, highly complex regulatory networks and multiple layers of regulatory pathways are interconnected to orchestrate the dynamic process of leaf senescence.

During the process of leaf senescence, leaves experience systematic physiological and biochemical changes, accompanied by alterations in gene expression. Genes that show increased expression are commonly referred to as *senescence-associated genes* (*SAGs*), while genes with decreased expression are known as *senescence down-regulated genes* (*SDGs*) [11,12]. The expression changes of numerous *SAGs* and *SDGs* contribute to the precise regulation of leaf senescence. Within the wide array of genes associated with senescence, the dynamic activation of transcription factors (TFs) is one of the hallmarks of leaf senescence [13,14]. These TFs present in multiple gene families, including NAC, WRKY, AP2/ERF, MYB, bZIP and bHLH, among which NAC and WRKY TFs have been widely studied and demonstrated to be essential in regulating senescence in different species [13,15,16,17,18]. NAC TFs plays both positive and negative roles in this process. In *Arabidopsis*, positive NAC regulators consist of ANAC082/ATAF1, ANAC016, ANAC019, ANAC028/AtNAP, ANAC032, ANAC046, ANAC055, ANAC059/ORS1, ANAC072, ANAC087 and ANAC092/ORE1 [19,20,21,22,23,24,25,26,27]. Intriguingly, only a limited number of NAC TFs exhibit negative regulatory functions, such as ANAC017, ANAC042/JUB1, ANAC075, ANAC082, ANAC083/VNI2, and ANAC090 [28,29,30,31]. WRKY family proteins are also one of the largest plant-specific TFs, characteristic for their invariant WRKYGQK motif [32]. It has been validated that WRKY6, WRKY22, WRKY26, WRKY42, WRKY45, WRKY53, WRKY54, WRKY55, WRKY70, WRKY71, and WRKY75 play vital roles in regulating leaf senescence in *Arabidopsis*, with WRKY54 and WRKY70 function as negative regulators [16,33,34,35,36,37,38,39,40].

Aside from NAC and WRKY, additional TFs such as the basic leucine zipper (bZIP) family, characterized by an alkaline region and a varied leucine zipper domain, also play an important role in the regulation of leaf senescence. However, only a small number of bZIP TFs have been found to be involved in regulation of leaf senescence. The bZIP TF G-box-binding factor 1 (GBF1/bZIP41), can bind to the G-boxes of the *CATALASE2* and *RBCS1a* promoter to inhibit their transcription, thereby triggering the initiation of leaf senescence [41]. ABA (abscisic acid)-responsive element (ABRE) binding factors/ABRE-binding proteins (ABFs/AREBs) ABF2, ABF3, ABF4 can directly bind to the promoter of Chlorophyll catabolic enzyme gene *NYE1*, *NYE2*, *PAO*, and *NYC1* to activate their transcription, thus promoting ABA-induced leaf senescence in *Arabidopsis* [42]. A recent study found that ABF2/3/4 could bind and transactivate the expression of the kinase kinase kinase 18 (MAPKKK18) coding gene, mediating ABA facilitated leaf senescence [43]. ABF3 and ABF4 can also activate the transcription of NAC TFs, such as ANAC072, ANAC055 and ANAC019, to promote leaf senescence, indicating that various transcriptional regulatory pathways are interconnected through complex and robust networks, allowing for fine-tuning of leaf senescence [44,45]. Further study on bZIP TFs associated with senescence could help shed light on the transcriptional regulatory mechanism underpinning leaf senescence regulation.

In the last few decades, there has been considerable advancement in elucidating the mechanism of leaf senescence in *Arabidopsis*, rice, wheat, tobacco and other species [15,17,46,47]. However, few studies have focused on the regulation of leaf senescence in legume forage. Alfalfa (*Medicago sativa*) is one of the most widely distributed forage legumes in the world because of its high yields, resistance, nutritional quality and adaptability [48,49,50]. At the same time, alfalfa has the characteristics of allogamy, self-incompatibly and autotetraploidy with a complex genetic background, impeding attempts to decipher the molecular principles of leaf senescence. Therefore, only a limit number of genes have been demonstrated to participate in leaf senescence regulation in alfalfa. For example, studies showed that alfalfa plants overexpressing the cytokinin biosynthetic gene *IPT* driven by the *SAG12* promoter, as well as those with a silenced chlorophyll catabolic gene which has been named STAY-GREEN (*SGR*) or overexpression of the γ-tocopherol methyltransferase gene *MsTMT*, exhibited a delayed leaf senescence phenotype, which could potentially improve the quantity and quality of forage [51,52,53]. In contrast, overexpressing *MsSAG113* could accelerate leaf senescence, indicating its crucial role in leaf senescence [54].

*M. truncatula*, a close relative of alfalfa, has been established as an appropriate model for genetic improvement and investigation of leaf senescence in forage crops due to its short growth period, self-pollination, small genome and high transformation efficiency [8,53,55,56]. Therefore, uncovering the molecular mechanism of leaf senescence in *M. truncatula* is of great theoretical and practical value for breeding high-yield varieties of alfalfa and other legume forage.

In order to determine the key regulators of leaf senescence, we employed a comparative transcriptome analysis over a time course, including young leaf, mature leaf and senescent leaf in *M. truncatula*. Further, we examined the detail expression profiles of leaf senescence and identified a novel bZIP TF MtbZIP60 that functions as a negative regulator of leaf senescence. This work will help to clarify the molecular regulatory mechanism underpinning leaf senescence in *M. truncatula* and provide a valuable gene resource for improving biomass yield and quality of legume forage.

## 2. Results

### 2.1. Changes in Physiological and Molecular Parameters during Age-Dependent Leaf Senescence

In order to comprehensively characterize the process of leaf senescence, we dynamically monitored the growth and senescence progression of *M. truncatula* ecotype R108 leaves. The content of chlorophyll and malondiadehyde (MDA) which are commonly used as indicators for the progression of leaf senescence were also detected at the same time, along with the expression of the senescence marker genes. For the purpose of consistency, the fifth and sixth compound leaf were utilized for investigation and subsequent transcriptome sequencing, and the day when the compound leaf was visible is considered as the first day. As illustrated in Figure 1, the expansion of leaves occurred at 5 d, reaching fully maturity at 15 d and senescence initiating at 30 d. Yellowing at the leaf tips became visible at 45 d, completely yellowing at 60 d and culminating with leaves falling (Figure 1A). The chlorophyll content rose from 3 d to 15 d, peaked at 15 d and then decreased (Figure 1B). The malondialdehyde (MDA) content displayed an intricate pattern throughout the process of leaf senescence. The MDA level remained consistent at 3 d and 5 d, experienced a slight decrease at 15 d, followed by an increase at 30 d and reached a maximum at 45 d and 60 d (Figure 1C). The expression of senescence up-regulated gene *MtORE1*, which is homologous to *Arabidopsis ANAC092/ORE1,* encoding a senescence-associated NAC TF, and senescence down-regulated gene *MtCAB1*, which is homologous to *Arabidopsis CAB1,* encoding a subunit of light-harvesting complex Ⅱ, changed correspondingly accompanied by leaf senescence (Figure 1D,E). Based on the results of phenotypic analysis and physiological data, we selected the leaves at 5 d, 30 d and 60 d, which represented young leaf, mature leaf and senescent leaf, respectively, to perform transcriptome sequencing.

### 2.2. Transcriptome Sequencing

Leaf samples at 5 d, 30 d and 60 d were harvested and subjected to transcriptome sequencing with three biological replicates at each point. A total of 54.79 Gb of clean data was collected, with each sample producing more than 5.75 Gb of data. The percentage of bases with a Phred value ≥ 30% (Q30) exceeded 89.74%, and the alignment efficiency of clean reads with the reference genome varied from 80.47% to 88.39% (Appendix A).

Principal component analysis (PCA) showed that the biological replicates from the same time point clustered together, indicating a significant level of similarity within the group. Across different stages of leaf senescence, the obtained transcriptomic data varied substantially, suggesting a higher degree of variability among different groups (Figure 2A). Additionally, the correlation between biological replicates was assessed using the Pearson correlation coefficient (PCC) analysis. The results revealed that all correlation coefficients between different samples within the same group exceeded 0.83, demonstrating a high level of consistency among the biological replicates (Figure 2B).

Using DESeq2 with screening criteria of adjusted *p*-value < 0.01 and |log_2_ (fold change)|>1 or <−1, the differentially expressed genes (DEGs) were identified in each group compared with one another (Appendix A). A comparation between 30 d and 5 d revealed a total of 9077 DEGs, consisting of 5095 down-regulated genes and 3982 up-regulated genes. A comparable number of DEGs was obtained between 60 d versus 5 d, with a total of 8321 DEGs, including 4760 down-regulated genes and 3562 up-regulated genes. Fewer DEGs were identified between 60 d versus 30 d, totaling 3511 DEGs, with 1763 down-regulated genes and 1748 up-regulated genes, suggesting a similar gene expression profile between 60 d and 30 d (Figure 2C). Following a comparative analysis of DEGs obtained from different combinations, it was found that 1057 DEGs were shared among the comparison (Figure 2D).

### 2.3. Validation of Transcriptome Data by qRT-PCR

In order to confirm the accuracy of RNA-seq results, quantitative real-time PCR (qRT-PCR) was performed on selected genes exhibiting different expression patterns in a biological independent experiment. A total of twelve candidate genes were selected, categorized into four distinct expression profiles: continuous up-regulation genes, continuous down-regulation genes, genes that were up-regulated at 30 d and then down-regulated at 60 d and genes that were down-regulated at 30 d, then up-regulated at 60 d. The results showed that the expression of the specific gene in RNA-seq was consistent with the one detected in qRT-PCR (Figure 3). The Pearson correlation coefficient was calculated by SPSS 17.0 software to evaluate the correlation between different detection methods. Overall, the results obtained from qRT-PCR were in line with the RNA-seq data, showing a correlation coefficient exceeding 0.82 (Figure 3). Thus, our data demonstrates the reliability and consistency of transcriptome analysis, which can be utilized for further investigating genes related to leaf senescence regulation in *M. truncatula*.

### 2.4. Short Time-Series Expression Miner (STEM) Clustering of DEGs during Leaf Growth and Senescence

In order to cluster and compare the obtained DEGs at different time points, we used Short Time-series Expression Miner (STEM) to identify the predominant model temporal expression profiles. Four significant and distinct expression profiles were identified, comprising three up-regulated profiles (14, 11 and 8) and one down-regulated profile (1) (Figure 4). Each profile consisted of a set of genes with a similar expression pattern.

### 2.5. Gene Ontology Enrichment and Kyoto Encyclopedia of Genes and Genomes Pathway Enrichment Analysis of Senescence-Associated Genes

To further identify genes associated with leaf senescence, Venn diagrams based on up-regulated DEGs obtained from comparisons between different time points were generated. A total of 108 *SAGs* shared by the three time points were detected (Figure 5A; Appendix A). Then, we performed GO enrichment analysis to characterize the gene function of the 108 *SAGs*. The analysis classified the gene function into three categories, including biological process, molecular function and cellular component. Specifically, the “cellular process”, “metabolic process”, “single-organism process”, “biological regulation” and “localization” were the main enriched terms in the biological process category. Significant enriched terms in the cellular component category included “cell”, “cell part”, “membrane”, “organelle”, “membrane part” and “organelle part”. In the molecular function category, enriched terms were “binding”, “catalytic activity”, “transporter activity” and “nucleic acid binding transcription factor activity” (Figure 5B; Appendix A).

Kyoto Encyclopedia of Genes and Genomes (KEGG) pathway enrichment analysis showed that the significant enriched pathway in the 108 *SAGs* were “endocytosis”, “indole alkaloid biosynthesis”, “spliceosome”, “protein processing in endoplasmic reticulum”, “nitrogen metabolism”, “beta-Alanine metabolism”, “alpha-Linolenic acid metabolism” and “RNA degradation” (Figure 5C; Appendix A).

### 2.6. Comparison of Senescence-Associated Genes between Medicago and Arabidopsis

As significant advancement had been made in understanding the molecular mechanism of leaf senescence in *Arabidopsis*, we compared the identified *SAG*s with homologous genes in *Arabidopsis* and analyzed their putative function in leaf senescence regulation using the leaf senescence database (LSD 5.0, https://ngdc.cncb.ac.cn/lsd/index.php (accessed on 1 May 2024)). In *Arabidopsis*, 92 out of 108 homologous genes are documented in the LSD database (Supplemental Appendix A). Notably, there are six genes in *Arabidopsis* known to play a role in senescence regulation, including *RD29B*, *SAUR72*, *AGL8*, *PYL5*, *WRKY71* and *LOX1* (Table 1). Additionally, two genes in *Arabidopsis* serve as marker genes for leaf senescence (Table 1). These findings confirm the validity of the experimental design and the accuracy of sampling timing.

### 2.7. Transcription Factor Analysis

Transcription factors (TFs) are recognized as key regulators of leaf senescence. We employed Plant Transcription Factor Database (PlantTFDB v5.0) to perform TF prediction. Among the 108 *SAGs* analyzed, 7 TFs were annotated by PlantTFDB 5.0, which belong to bHLH (two genes), NF-X1, bZIP, MADS and WRKY (two genes) TF families (Figure 6A; Appendix A). One of the identified WRKY TFs, homologs to WRKY71 in *Arabidopsis*, has been demonstrated to positively regulate leaf senescence. Nevertheless, there is currently no report about whether the remaining six genes also play a role in the regulation of leaf senescence, thus requiring further investigation.

### 2.8. MtbZIP60 Functions as a Novel Regulator of Leaf Senescence

As only a limited number of bZIP TFs have been identified as regulators of leaf senescence in different species, the identified bZIP TF MtbZIP60 was chosen to perform further functional analysis. We first validated the expression trend of MtbZIP60 during leaf growth and senescence. The result showed a significant increase in *MtbZIP60* expression level as leaf aging (Figure 6B, Appendix A). In order to investigate the roles of MtbZIP60, we obtained a *Tnt1*-inserted mutant line *mtbzip60* through screening the mutant collection. The *MtbZIP60* gene contains a single exon, with the *Tnt1* insertion at 16 bp downstream of the translational start codon (Figure 6C). qRT-PCR analysis showed that the insertion of *Tnt1* leads to the loss of full-length expression of *MtbZIP60* (Figure 6D). Dark treatment is an effective method to induce leaf senescence and is commonly used to simulate synchronous senescence [15]. Therefore, the fifth fully expanded compound leaf from apex was detached for dark treatment. After a five-day dark treatment, the detached leaves of *mtbzip60* showed an accelerated leaf senescence compared with the wild-type R108 (Figure 6E). The degradation of chlorophyll, as reflected by the SPAD value, was significant higher in *mtbzip60* compared with R108, which is consistent with the differences observed in the senescence process (Figure 6F). In addition, the senescence marker genes, *MtORE1* and *MtCAB1*, exhibited corresponding alterations as well (Figure 6G,H). These results suggested that MtbZIP60 acts as a novel regulator of leaf senescence in *M. truncatula*.

### 2.9. Expression Pattern, Subcellular Localization and Transcriptional Activity Analysis of MtbZIP60

qRT-PCR was performed to analyze the temporal and spatial expression of *MtbZIP60*. The result revealed that *MtbZIP60* exhibited relatively high-level expression in leaf, moderate-level expression in root and stem and low-level expression in pod and seed (Figure 7A). The observed peak expression in leaf is consistent with its regulatory role in leaf senescence.

To determine the subcellular localization of MtbZIP60, green fluorescent protein (GFP) was fused to the C-terminus of MtbZIP60 and expressed in tobacco epidermal cells, driven by the cauliflower mosaic virus 35S promoter. In contrast to the GFP control, where the fluorescence was detectable in both the cytoplasm and nucleus, the MtbZIP-GFP fluorescence specifically localized to the nucleus (Figure 7B).

To analyze the transcriptional activity of MtbZIP60, the CDS of *MtbZIP60* was cloned into *pGBKT7* vector, to generate *GAL4BD-MtbZIP60* fusion. The *pGBKT7* vector and *GAL4BD-TaNAC6* were used as negative and positive controls, respectively [15]. Compared to the negative control, *GAL4BD-MtbZIP60* and *GAL4BD-TaNAC6* could activate the expression of reporter genes in yeast, as evidenced by normal growth on the selective SD medium lacking Trp, His and Ade (Figure 7C). The result indicates a potential involvement of MtbZIP60 in transcriptional regulation.

### 2.10. MtbZIP60 Interacts with MtWRKY40 in Regulation of Leaf Senescence

To elucidate the regulatory mechanism of MtbZIP60 in regulation leaf senescence, we conducted a yeast two-hybrid library screening to identify MtbZIP60-interacting proteins. The CDS of *MtbZIP60* was cloned into *pGBKT7*, serving as a bait. The selective medium was supplemented with different concentrations of 3-AT to evaluate its inhibitory effect on the transactivation activity of MtbZIP60. The result showed that a concentration of 60 mM of 3-AT could completely inhibit the transcriptional activation activity of MtbZIP60 and be used for Y2H screening. A total of 72 clones were obtained and sequenced, out of which three copies were identified as a WRKY TF, MtWRKY40 (Medtr2g105060). We then amplified the CDS of *MtWRKY40* and *MtbZIP60* and constructed them into *pGBKT7* vector and *pGADT7* vector, respectively, for Y2H assay. The result showed that the yeast co-transformed with BD-MtWRKY40 and AD-MtbZIP60 could grow on SD/-Trp-His-Ade + X-α-gal medium, confirming the MtbZIP60-MtWRKY40 interaction in yeast (Figure 8A). The interaction was verified via a luciferase complementation imaging (LCI) assay in *Nicotiana benthamiana* using split luciferase protein. Strong luciferase (LUC) signal was detected in tobacco leaf between MtbZIP60-nLUC and cLUC-MtWRKY40, whereas no LUC signals were observed in the negative control (Figure 8B).

To determine the potential role of MtWRKY40 in leaf senescence, we analyzed its expression trend during leaf growth and senescence, and the result showed a significant up-regulation in the 60-day-old senescent leaves, indicating MtWRKY40 may participate in the regulation of leaf senescence (Figure 8C). These results suggest MtbZIP60 inhibits leaf senescence through its interaction with MtWRKY40.

## 3. Discussion

Leaf senescence is a crucial issue in leaf development, closely related to the biomass production and quality of forage crops. Premature leaf senescence can lead to significant biomass loss and decreased nutritional quality, whereas delaying leaf senescence may serve as a feasible strategy to improve forage biomass production and quality. *M. truncatula* is a close relative of alfalfa. Investigating the regulatory mechanism behind leaf senescence in *M. truncatula* can provide valuable insights into how leaf senescence is controlled or could directly indicate the function of homologues genes in alfalfa. This is exemplified by the functional analysis of *SGR* in alfalfa, which is initially cloned from *M. truncatula*, with both performing a similar function in chlorophyll degradation [53]. Transcriptomic analyses are an efficient and reliable approach for identifying DEGs and have been utilized to screen for underling *SAGs* and TFs involved in regulating leaf senescence in various species, including wheat, rice, maize, foxtail millet, cotton, sorghum and *Arabidopsis*, which provide valuable insights into the mechanisms that govern this complicated biological process [63,64,65,66,67].

In this study, we examined age-dependent leaf senescence in *M. truncatula* and employed transcriptome analysis to investigate *SAGs*. To determine the optimal timing for sampling, we continuously monitored the growth and senescence process of leaves. During progression of senescence, malondialdehyde (MDA) and chlorophyll content are commonly utilized as indicators due to their noticeable alteration. Our data indicated that the chlorophyll content in leaves, as reflected by the SPAD value, peaked at 5 days after emergence and exhibited a significant downward trend from 30 to 60 days. Leaf yellowing became visible at 45 days and particularly prominent at 60 days, which corresponds to the chlorophyll content and the progression of leaf senescence (Figure 1A,B). The MDA content exhibited a decrease at 15 days and significant increase at 45 days and 60 days, consistent with the progression of leaf senescence. Our data suggested that the specific time points, 5 day, 30 day and 60 day, selected for transcriptomic analysis corresponded to the stages of the young leaf, mature leaf and senescent leaf, respectively.

As senescence advances, the expression of genes is closed linked to metabolic processes, including those related to the degradation of chlorophyl, amino acids, lipids, carbohydrates and nitrogen compounds [7,53,63]. In our study, we identified 108 genes that continuously up-regulated genes during leaf development and senescence, which appear to be involved in multiple regulatory pathways in leaf senescence. GO and KEGG enrichment analyses revealed that the identified *SAG*s were primarily enriched in nitrogen metabolism, amino acid metabolism, RNA degradation, linolenic acid metabolism and glutathione metabolism (Figure 4). These results provide insights into the physiological and biochemical changes that occur during leaf senescence in *M. truncatula*.

Activation of transcription factors (TFs) are one of the hallmarks during the initiation and progression of leaf senescence [13]. Over the past decades, considerable progress has been made in our comprehension of the TFs’ functions in the regulation of leaf senescence [13,15,17]. Transcriptome analysis is also used for analysis and mining of TFs implicated in this process [7,56]. In our study, we predicted TFs within the obtained *SAGs* through the PlantTFDB database and found that TFs from the WRKY, bHLH, NF-X1 and bZIP families were identified, which play a role in senescence regulation (Figure 5A). Medtr5g091390, one of the predicted WRKY TFs, is homologous to WRKY71, which has been demonstrated to positively regulate leaf senescence via multiple pathways in *Arabidopsis thaliana* [37]. Thus, there are reasons to believe that Medtr5g091390 may also participate in the regulation of leaf senescence, even though its function is currently ambiguous and warrants further investigation. Other TFs, MTR_2g038040 (bHLH), MTR_4g027040 (NF-X1), MTR_4g131160 (bHLH) and MTR_4g122530 (WRKY), were continuously up-regulated during leaf senescence yet have not been proved to be directly involved in the regulation of leaf senescence and deserve further investigation.

WRKY and NAC are the most extensively studied TF families that regulate leaf senescence [13,14]. Additionally, the bZIP family is also essential in this biological process, although fewer members have been cloned [41]. Through transcriptome analysis, a novel bZIP TF MtbZIP60 was identified as exhibiting a continuously up-regulated expression pattern throughout leaf growth and senescence. Phenotypic analysis of the *Tnt1* retrotransposon-tagged mutant line *mtbzip60*, which lacks the full-length transcript, demonstrated a delayed leaf senescence phenotype, suggesting that MtbZIP60 acts as a negative regulator of leaf senescence (Figure 5). Additional phenotypic evaluation of *MtbZIP60* overexpression in suppressing leaf senescence will provide more solid evidence to validate its negative impact on leaf senescence regulation. Subcellular localization analysis revealed MtbZIP60 is localized in the nucleus and exhibits transcriptional activation activity, suggesting its protentional involvement in transcriptional regulation (Figure 7). We also found that MtbZIP60 can associate with MtWRKY40, which also showed an up-regulated pattern during leaf senescence (Figure 7), whereas the precise role of MtWRKY40 in leaf senescence regulation requires additional investigation. Further research on MtbZIP60 and its interaction with MtWRKY40 in the regulation of leaf senescence in *M. truncatula* will enhance our understanding of the transcriptional regulation mechanism underlying leaf senescence.

Leaf senescence is pivotal for plant growth, stress tolerance, reproductivity and fitness. In crops, the timing of leaf senescence profoundly influences grain yield and nutritional quality, making it a key agronomic trait [68]. In forage crops where nutrients are primarily stored in leaves, leaf senescence may directly affect the overall forage quality [8]. Alfalfa is one of the most important legume forage species in the world due to its high yield, nutritional quality and adaptability. It is believed that preventing premature leaf senescence and cultivation of new varieties of alfalfa with optimized leaf senescence are effective ways to improve the quantity and quality of alfalfa [7,8,51]. In our study, we conducted a systematical analysis of the dynamic gene expression pattern during leaf senescence in *M. truncatula*, a close relative of alfalfa, and obtained a series of *SAGs* in which MtbZIP60 was identified as a positive regulator. Identification of homologous genes to *MtbZIP60* in alfalfa and investigation of its precise function in regulating leaf senescence are essential. Further evaluating its practical influence on forage yield and quality under field conditions could not only help our comprehension of the mechanism involved in this process in alfalfa, but it may also provide valuable gene resources and practical methods for cultivating new alfalfa varieties with manipulated leaf senescence processes.

## 4. Material and Methods

### 4.1. Plant Materials and Growth Condition

The *M. truncatula* ecotype R108 was used in this study. NF4038 (*mtbzip60*) was obtained by screening tobacco retrotransposon *Tnt1*-tagged mutant collection of *M. truncatula*. For germination, seeds were sown to petri dishes with moistened filter paper after imbibition, followed by a 7-day stratification at 4 °C. Subsequently, the dishes were then moved to a growth chamber overnight to promote seed germination at 24 °C. The germinated seeds were then grown in the greenhouse under the following conditions: 24 °C day/20 °C night temperature, 16-h day/8-h night photoperiod and 60% to 70% relative humidity. The fifth compound leaves were harvested at 5 d, 30 d and 60 d after emerging and frozen in liquid nitrogen. Three biological repeats were prepared, each comprising individual leaf from five plants.

### 4.2. Determination of Chlorophylls and MDA Content

The determination of chlorophyll content was carried out using a SPAD 502 Plus Chlorophyll Meter (Konica Minolta, Tokyo, Japan), with reads indicated as SPAD values.

The measurement of malondialdehyde (MDA) content was performed through thiobarbituric acid (TBA) reaction, as described by Saher [69]. Briefly, 0.5 g of leaf tissue was homogenized by grinding in liquid nitrogen, followed by adding 10 mL 0.1% trichloroacetic acid (TCA) and vortex mixing. The homogenate was then centrifuged at 12,000 rpm for 5 min and 4 mL 20% TCA containing 0.5% TBA were add to 1 mL of the supernatant. The mixture was heated and rapidly cooled down in an ice slurry. After centrifugation, the absorbance of the solution was measured at wavelengths of 450 nm, 532 nm and 600 nm. The concentration of MDA was calculated using an extinction coefficient of 155 mM^−1^ cm^−1^.

### 4.3. RNA Isolation and Gene Expression Analysis

Total RNA was extracted Using TRIzol^®^ Reagent (TAKARA, Cat#9109, Kusatsu, Shiga, Japan). First strand cDNA was synthesized from 1 μg of total RNA using HiScript^®^ II QRT Super-Mix (Vazyme, R222-01, Nanjing, China) according to the manufacturer’s instructions. RT-PCR was performed at 98 °C for 5 min, followed by 28–40 cycles of amplification (98 °C for 20 s, 55–60 °C for 20 s and 72 °C for 20 s). Quantitative RT-PCR was performed on a CFX96 Real-Time System (Bio-Rad, Hercules, CA, USA) with ChamQ SYBR qPCR Master Mix (Vazyme, Q711-02). The qRT-PCR reaction consists of 10.0 μL of 2 × ChamQ Universal SYBR qPCR Master Mix, along with 0.4 μL each of forward and reverse primers, and template cDNA, reaching a total volume of 20 μL. Relative gene expression levels were calculated using the 2^−ΔΔCt^ method with a *M. truncatula ACTIN* gene as internal control. The qRT-PCR were performed with at least three biological replicates, each containing four technical replicates. qRT-PCR primers were designed using Primer3web version 4.1.0. All primers used in this study are listed in Appendix A.

### 4.4. Library Construction and Transcriptome Sequencing

Total RNA was extracted from leaves at different developmental stages. RNA concentration and purity was analyzed using NanoDrop 2000 (Thermo Fisher Scientific, Wilmington, DE, USA). RNA integrity was assessed using the RNA Nano 6000 Assay Kit of the Agilent Bioanalyzer 2100 system (Agilent Technologies, Santa Clara, CA, USA). RNA libraries were generated using the NEBNext^®^ UltraTM RNA Library Prep Kit for Illumina^®^ (NEB, Ipswich, MA, USA) based on manufacturer’s guidelines. In brief, mRNA was purified using oligo (dT) beads and fragmented with fragmentation buffer. Double-stranded cDNA was obtained using a double-stranded cDNA synthesis kit (NEB, Ipswich, MA, USA). After an “A” base was added to the 3′ ends of DNA fragments, NEBNext Adaptor with hairpin loop structure were ligated. Then, the library fragments were purified with AMPure XP system (Beckman Coulter, Beverly, CA, USA) to select for ~240 bp cDNA fragments, followed by PCR amplified using Phusion High-Fidelity DNA polymerase (NEB, Ipswich, MA, USA). After purification of PCR products, library quality was assessed on the Agilent Bioanalyzer 2100 system. Clustering was performed on a cBot Cluster Generation System using TruSeq PE Cluster Kit v4-cBot-HS (Illumia), and RNA-seq sequencing library was carried out on Illumina NovaSeq 6000 platform.

The raw reads were processing through in-house Perl scripts, and obtained clean reads were mapped to *M. truncatula* reference genome (http://plants.ensembl.org/Medicago_truncatula/Info/Index (accessed on 1 March 2022) using Hisat2 2.0.4 (http://ccb.jhu.edu/software/hisat2/index.shtml (accessed on 20 March 2022)). The gene expression level was indicated as cDNA fragments per kilobase per million fragments that mapped to the *M. truncatula* reference genome. Differentially expressed genes (DEGs) between different samples were obtained using DESeq2 1.6.3 with adjusted *p*-value < 0.01 and |log_2_ (fold change)|> 1 or <−1.

### 4.5. Transcription Factor Analysis

The Plant Transcription Factor Database (PlantTFDB 5.0, https://planttfdb.gao-lab.org/prediction.php (accessed on 5 May 2022)) was used with the default parameters to perform TF prediction according to the domain information contained.

### 4.6. Gene Ontology and Kyto Encyclopedia of Genes and Genomes Pathway Enrichment Analysis

The Gene Ontology (GO) and Kyto Encyclopedia of Genes and Genomes (KEGG) pathway enrichment analyses were performed on the platform BMKCloud (www.biocloud.net (accessed on 25 March 2022)). Specifically, GO enrichment analysis was implemented by the GOseq R packages (3.19)-based Wallenius non-central hyper-geometric distribution (accessed on 5 April 2022) [70], which can adjust for gene length bias in DEGs. KOBAS 2.0 [71] software (accessed on 10 April 2022) was used to test the statistical enrichment of differential expression genes in KEGG pathways.

### 4.7. Protein Subcellular Localization

The coding sequence of *MtbZIP60* was amplified and introduced into Gateway binary vector *pMDC83*. Construct of GFP was used as positive control and mRFP-AHL22 as nuclear localized marker. The obtained vector was transformed into *Agrobacterium tumefaciens* strain GV2260 and then used to infiltrate tobacco leaves in the infiltration buffer, consisting of 10 mM 2-morpholinoethanesulphonic acid (MES, pH = 5.8), 200 μM acetosyringone and 10 mM MgCl_2_. The final OD_600_ of each recombinant *Agrobacterium tumefaciens* strain was 0.5. Abaxial sides of the 4th and 5th leaves of 4-week-old tobacco (*N. benthamiana*) leaves from the top were selected and infiltrated with the *Agrobacterium tumefaciens* mixture. The infiltrated tobaccos were grown in the dark for 12 h before being transferred to the greenhouse, where they were sampled after 3 days for investigation. GFP-MtbZIP60 signal was examined using Olympus FV3000 confocal microscopy.

### 4.8. Transactivation Activity Assay in Yeast

The coding sequence of *MtbZIP60* was cloned into *pGBKT7* to generate *GAL4BD-MtbZIP60* fusion. The resulting constructs and positive control *GAL4B*D-*TaNAC6* were transformed into *Saccharomyces cerevisiae* strain AH109, respectively. The transformed yeast was then screened on SD/-Trp medium. Positive recombinant clones were resuspended with sterilized water and inoculated on SD/-Trp-His-Ade + X-α-gal for analysis of transactivation activity.

### 4.9. Yeast Two-Hybrid (Y2H) Interactions and Library Screening

The Y2H cDNA library was constructed by OE biotech (Shanghai, China) via cloning the full-length cDNA library prepared from different tissues into the *pGADT7* vector and transformed into yeast Y187 strain. The coding sequence of *MtbZIP60* was inserted into the *pGBKT7* vector, fusing with GAL4BD. The resulting construct was transformed into yeast strain AH109 and mated with the cDNA library. After mating, cultures were spread on 35 large Petri dishes containing SD/-Ade-His-Leu-Trp supplied with 60 mM of 3-AT to inhibit the transactivation activity of MtbZIP60 and 20 μg/mL X-α-gal for blue colony selection and incubated for 3–5 days at 28 °C. The coding sequences of identified candidates were amplified, sequenced and cloned for further validation.

For pairwise Y2H assay, it was performed as previous described [72]. The full-length CDSs of candidates were amplified and inserted into the *pGADT7* vector. Then, the prey and bait plasmids were co-transformed into yeast strain AH109 and incubated on SD/-Leu-Trp and SD/-Ade-His-Leu-Trp selective media to test for interaction.

### 4.10. Luciferase Complementation Imaging Assay (LCI)

The LCI assay was performed as previous described [73]. The CDSs of *MtbZIP60* and *MtWRKY40* were amplified and used to subclone the different constructs (*MtbZIP60-nLuc*, *cLuc-MtWRKY40*). *Agrobacterium* cells (GV3101) containing different pair of plasmids were adjusted to optical density (OD_0.5–0.6_) with infiltration buffer and infiltrated into tobacco leaves. After incubation in the dark for 12 h and normal culture for 3 days, the luciferase substrates (D-luciferin potassium salt, LUCK, GOLDBIO) were sprayed onto the infiltrated leaves and further analyzed by a low-light cooled charge-coupled imaging device (VILBER, FUSION FX7 Spectra).

### 4.11. Statistical Analysis

Unless particularly stated, statistical significance test was conducted by a two-tailed Student’s *t*-test. Differences were considered statistically significant when *p* ≤ 0.05. * and ** indicate *p* < 0.05 and *p* < 0.01, respectively.

### 4.12. Accession Number

The gene sequence can be accessed via Phytozome (https://phytozome-next.jgi.doe.gov/ (accessed on 7 September 2022)) database under the following accession numbers: *MtbZIP60* (Medtr4g070860), *MtWRKY40* (Medtr2g105060), *MtORE1* (Medtr4g108760), *MtCAB1* (Medtr4g094605), which are based on ecotype A17 genome sequence.

## 5. Conclusions

In summary, we carried out a comparative transcriptome analysis using young, mature and senescent leaves of *M. truncatula*. A total of 1057 DEGs were obtained, in which 108 genes exhibited a consistently up-regulated pattern during leaf senescence. Seven TFs were identified within the 108 *SAG*s, and a novel bZIP TF MtbZIP60 was demonstrated to negatively regulate leaf senescence through interacting with MtWRKY40. Our results help shed light on the transcriptional regulation mechanisms underlying leaf senescence in *M. truncatula* and provide potential targets for manipulating leaf senescence progression to improve the biomass yield and quality of alfalfa.

## Figures and Tables

**Figure 1 ijms-25-10410-f001:**
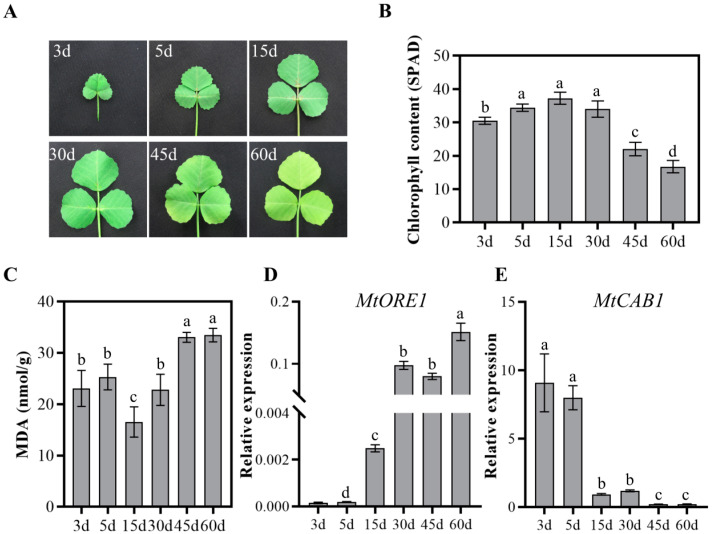
Physiological and biochemical analysis of leaf growth and senescence in *Medicago truncatula*. (**A**) Phenotype of the fifth compound leaf at different growth and senescence stages. The day when the fifth compound leaf emerged is considered as day 1. (**B**) Chlorophyll content in the fifth compound leaf at 3 d, 5 d, 15 d, 30 d, 45 d and 60 d. The chlorophyll content was obtained by measuring the SPAD value using a chlorophyll meter. Date is shown as mean ± SD (*n* = 4). Significant differences revealed by Tukey’s multiple comparison test are indicated by letters above bars (*p* < 0.05). (**C**) MDA content was measured in the fifth compound leaf at indicated time points. Date is shown as mean ± SD (*n* = 4). Significant differences revealed by Tukey’s multiple comparison test are indicated by letters above bars (*p* < 0.05). (**D**,**E**) Expression of senescence up-regulated and down-regulated marker genes *MtORE1* (**D**) and *MtCAB1* (**E**) at 3 d, 5 d, 15 d, 30 d, 45 d and 60 d. Date is shown as mean ± SD (*n* = 3). Significant differences revealed by Tukey’s multiple comparison test are indicated by letters above bars (*p* < 0.05).

**Figure 2 ijms-25-10410-f002:**
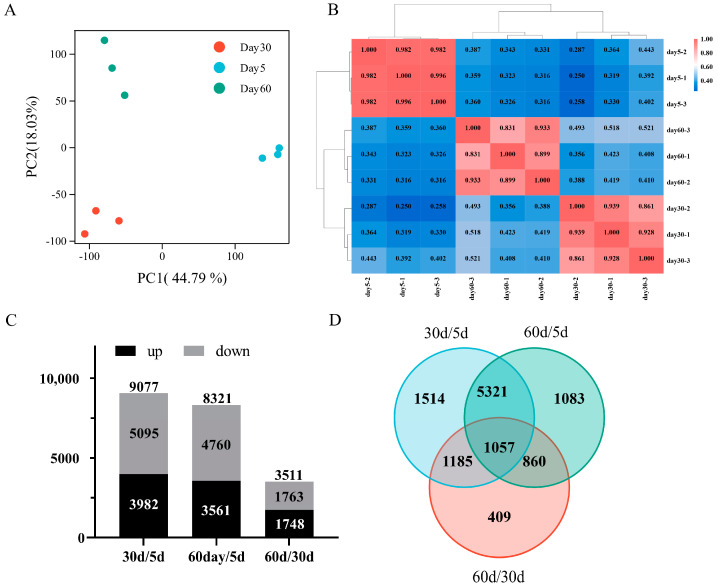
Transcriptomic overview of age-dependent leaf senescence in *Medicago truncatula*. (**A**) Principal component analysis (PCA) plot of transcriptome data from different leaf growth and senescence stages; (**B**) Pearson correlation coefficient of transcriptome profiles from different leaf growth and senescence stages; (**C**) The number of differentially expressed genes (DEGs) identified from various comparison combination, as determined based on FPKM values using DESeq2 with adjusted *p*-value < 0.01 and |log_2_ (fold change)|>1 or <−1; (**D**) Venn diagram of overlap DEGs among different comparisons.

**Figure 3 ijms-25-10410-f003:**
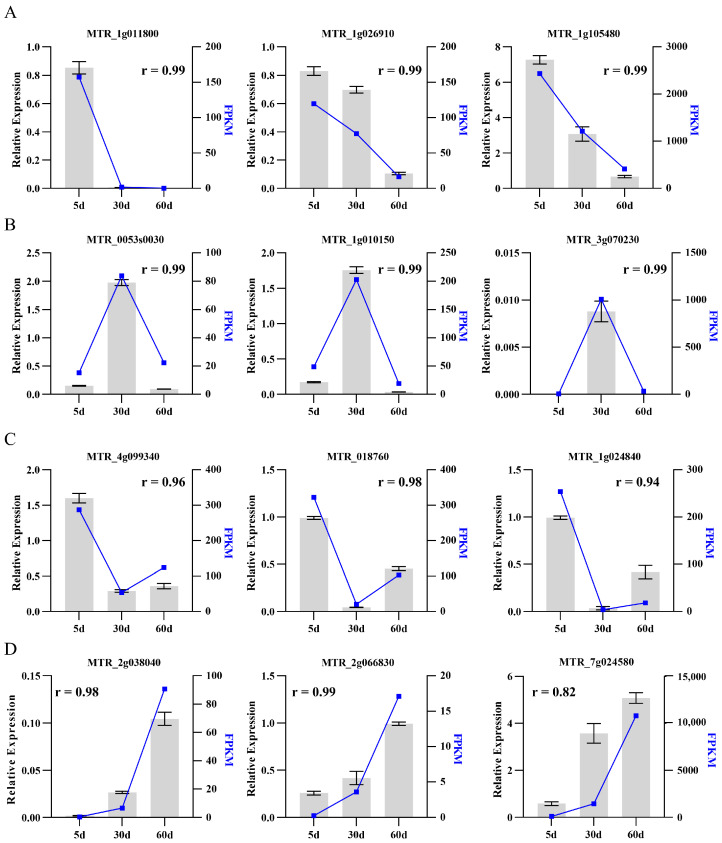
(**A**–**D**) Validation of selected transcripts by qRT-PCR. The expression of the specific gene in RNA-seq was represented by the lines on the right y-axis, whereas the relative expression of the same gene detected by qRT-PCR from independent replicates was depicted by the columns on the left y-axis. Date is shown as mean ± SD (*n* = 3).

**Figure 4 ijms-25-10410-f004:**
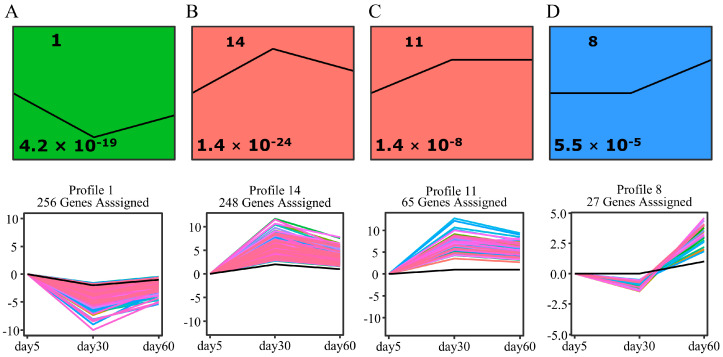
Short Time-series Expression Miner (STEM) clustering of 1057 DEGs shared by the three time points. (**A**–**D**) indicate different types of model expression profile, with the model profile ID number on the top left-hand corner and *p*-value on the bottom left. Different colors represent different significance. Only the model temporal expression profiles that has a significant number of assigned genes compared to the number of expected genes, with a *p*-value < 0.05, are displayed.

**Figure 5 ijms-25-10410-f005:**
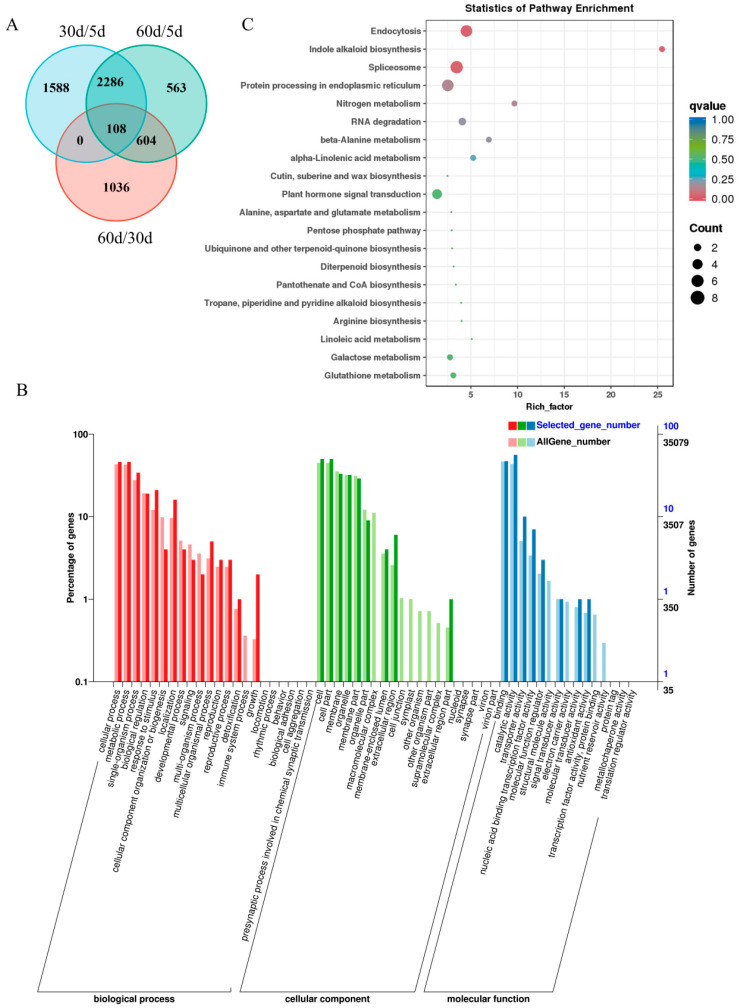
Comparison of *SAGs* among different comparison groups and Gene Ontology (GO) enrichment and Kyoto Encyclopedia of Genes and Genomes (KEGG) enrichment analysis. (**A**) Venn diagram of *SAGs* in different comparison groups. (**B**) The GO enrichment analysis of the 108 *SAGs* shared by all the comparison group. The X-axis indicates different GO terms. The left Y-axis represents the percentage of genes, and the right Y-axis represents the number of genes enriched for the relevant GO terms. (**C**) KEGG pathway enrichment analysis of the 108 *SAGs* shared by all the comparison group. The X-axis indicates enrichment factor, which represents the ratio of the proportion of genes annotated to a specific pathway among DEGs to the proportion of genes annotated to the same pathway among all genes. Dot color indicates *q*-value, which is the *p*-value corrected by multiple hypotheses testing.

**Figure 6 ijms-25-10410-f006:**
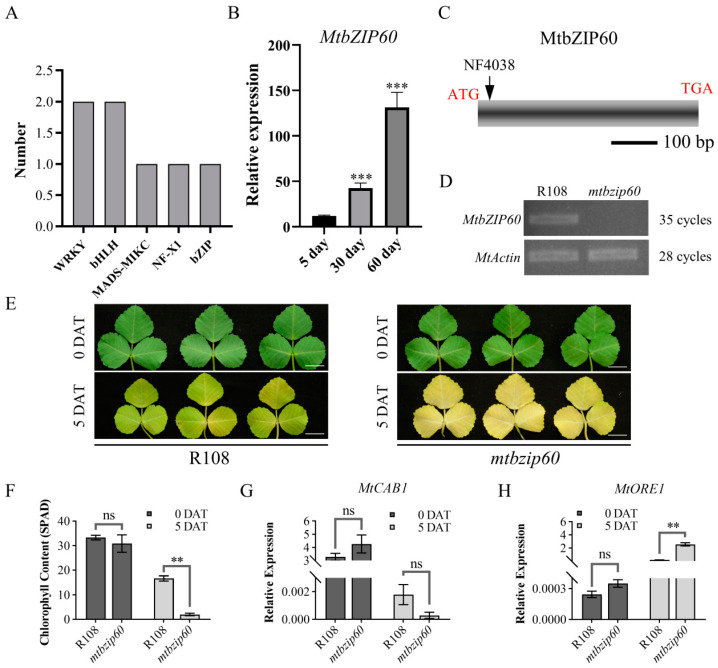
MtbZIP60 functions as negative regulator of leaf senescence. (**A**) Prediction of TFs within the 108 *SAGs*. (**B**) Expression trend of *MtbZIP60* during leaf growth and senescence. (**C**) Schematic representation of the structure of *MtbZIP60* and the position of *Tnt1* in *mtbzip60*. (**D**) RT-PCR analysis of *MtbZIP60* in wild type (R108) and *mtbzip60*. (**E**) Detached leaves of WT and *mtbzip60* in the dark for 5 days. Bars = 1 cm. (**F**) Measurement of chlorophyll content (SPAD) in (**E**). (**G**,**H**) qRT-PCR detection of *MtCAB1* (**G**) and *MtORE1* (**H**) transcriptional level in (**E**). DAT, days after treatment. Date is shown as mean ± SD. Asterisks indicate significant difference from the WT (Student’s *t* test, ** *p* < 0.01, *** *p* < 0.001, ns: no significant). At least three biological replicates were performed.

**Figure 7 ijms-25-10410-f007:**
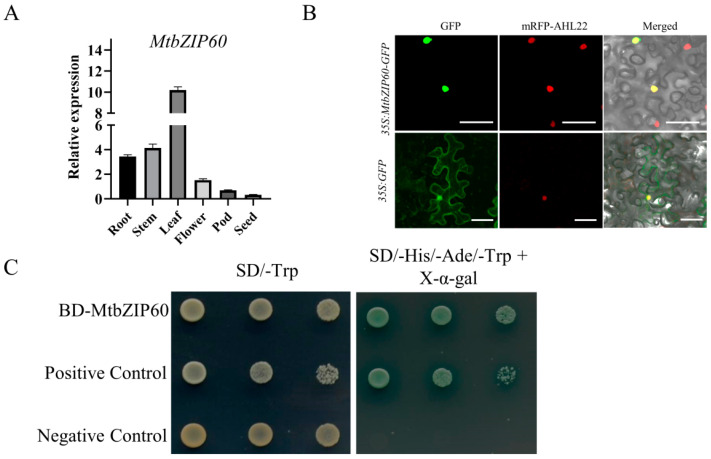
Expression pattern, subcellular localization and transcriptional activity analysis of MtbZIP60. (**A**) Expression pattern of *MtbZIP60* in different tissues detected using qRT-PCR. *MtActin* was used as an internal control. (**B**) Subcellular localization of GFP and MtbZIP60-GFP in epidermal cells of tobacco leaf. The mRFP-AHL22 fusion was used as nuclear localization marker. Green represents the green fluorescent protein, red represents the red fluorescent protein, and yellow is obtained by combining the two fluorescent proteins. Bars = 20 μm. (**C**) Transactivation activity analysis of MtbZIP60 in yeast. Full-length CDS of *MtbZIP60* and *TaNAC6* were cloned into the pGBKT7 vector and transformed into yeast strain AH109. Transformants harboring different plasmids were inoculated onto selective medium, with BD-TaNAC6 serving as the positive control.

**Figure 8 ijms-25-10410-f008:**
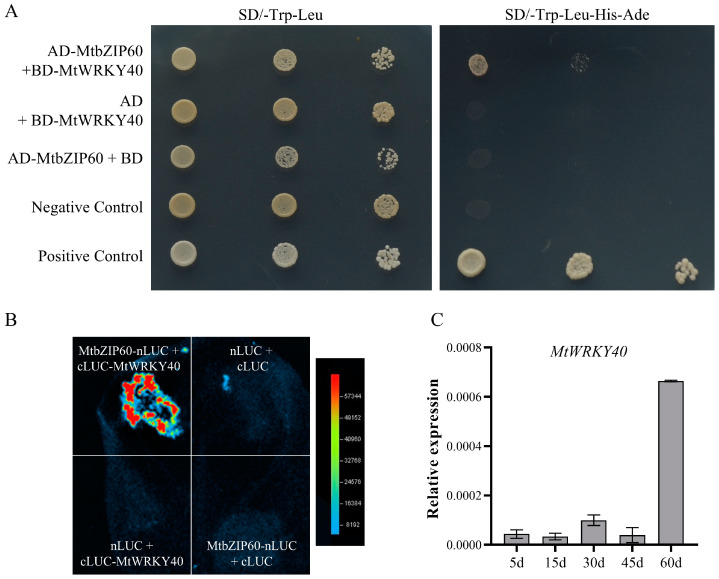
MtbZIP60 physically interacts with MtWRKY40 in the regulation of leaf senescence. (**A**) Interaction of MtbZIP60 and MtWRKY40 in an Y2H assay. Yeast cells were co-transformed with AD-MtbZIP60 and BD-MtWRKY40 and cultured on selective medium. Auxotrophic growth on the SD/-Trp-Leu-His-Ade medium indicates the interaction. (**B**) Interaction of MtbZIP60 and MtWRKY40 in *N.benthamiana* leaf using an LCI assay. MtbZIP60 was fused with the n-terminal of LUC to generate MtbZIP60-nLUC. MtWRKY40 was fused with the c-terminal of LUC to generate cLUC-MtWRKY40. Different pairs of constructs were used for infiltrating tobacco leaves. (**C**) Expression trend of *MtWRKY40* during leaf growth and senescence.

**Table 1 ijms-25-10410-t001:** Homologous genes involved in senescence regulation in *Arabidopsis.*

Number	*Medicago truncatula* Gene Id	Homologous Gene Id in *Arabidopsis thaliana*	*Arabidopsis thaliana* Gene Name	Effect for Senescence	Reference
1	Medtr2g066830	AT3G12830	*SAUR72*	Promote	[57]
2	Medtr1g100623	AT5G52300	*RD29B*	Delay	[30]
3	Medtr4g109830	AT5G60910	*AGL8*	Promote	[58,59]
4	Medtr5g083270	AT5G05440	*PYL5*	Promote	[60]
5	Medtr5g091390	AT1G29860	*WRKY71*	Promote	[37]
6	Medtr8g018730	AT1G55020	*LOX1*	Promote	[61]
7	Medtr8g074270	AT3G51430	*YLS2*	Marker gene	[62]

## Data Availability

The datasets present in this study can be found within the article and its supplemental data. The transcriptome raw data can be found at the NCBI database with the project ID1131009 (http://www.ncbi.nlm.nih.gov/bioproject/1131009 (accessed on 23 September 2024)).

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
