# Peer review of "Identification of a Novel Gene MtbZIP60 as a Negative Regulator of Leaf Senescence through Transcriptome Analysis in Medicago truncatula"

_ijms, 2024, doi:10.3390/ijms251910410_

Round 1

Reviewer 1 Report

Comments and Suggestions for Authors

This paper presents new results related to gene expression changes associated with leaf senescence in a study model closely related to the crop alfalfa. The paper then focuses on one candidate transcription factor for this trait, confirming its association with leaf senescence and identifying interactions with other transcription factors. The abstract provides a clear summary but the statements that genes have been proved to be involved in leaf senescence should be justified by stating that you present additional tests of function. A clear overview of the topic with an appropriate level of detail is provided in the introduction. The aims and approach are described simply at the end of this section but more detail is needed about the follow-up tests of the novel bZIP gene you identify. The results are well organised and summarised with figures and tables. Later parts of the results focus only on the up-regulated gene results. It is important that you justify why you did not focus on down-regulated genes in the same way. Subsequent functional validation experiments are explained well with a logical sequence explaining how each experiment led to the next. The discussion contains some repetition from the introduction that could be cut if necessary but it also places the results of this study into wider context. Recommended further study of other identified TFs are highlighted, that provide useful future research directions for the topic. The discussion ends with some nice overview about the wider importance of the topic to agriculture. The methods are mostly appropriate but some essential extra details should be added as described in the specific comments.

Specific comments

L26 This statement about having proved that MtbZIP60 inhibits leaf senescence requires that you briefly mention how you performed follow-up experiments to test function. 

L81 Again, I would be careful about using the verb "proved" if these papers simply report associated gene expression differences.

L95 Spell out ABA at first use, to highlight this hormone to readers.

L112 Replaces "grasses" with "legumes" to avoid taxonomic confusion. Specify the kinds of resistance that alfalfa possesses. 

L132 Spell out MDA at first use.

L178-179 "Down-regulated" repeated twice. One of these should be "up-regulated"

Figure 3. Is it possible to show also the error bars for qPCR data? Perhaps qPCR points could be offset slight to the side of RNAseq bars to show these error bars more clearly.

L228-229 Justify focusing only on SAGs and not SDGs from this point in the analysis. Why do you expect up-regulated genes to be more important for senescence than down-regulated genes?

Table 1 Gene name YLS2 repeated twice.

L287-289 This justification for follow-up studies of MtbZIP60 could also be added to the aims section at the end of the introduction.

L415-416 This statement seems to have errors. The figure should be Figure 6. The results shown in this figure suggest that the mtbzip60 mutant exhibits accelerated leaf senescence.

L442-444 Please confirm that the parent like of the mutant collection was also R108 or justify using a different line for this collection.

L445 Cold treatment before germination is better referred to as "stratification" rather than "vernalization".

L445-446 State the seed germination temperature.

L473 Please include the RT-PCR annealing temperature and amplification cycle number for each primer pair listed in Table S1.

L478-479 Provide extra details about the read mapping and any initial quality control steps.

L491 Briefly describe the amplification conditions and products used. 

L492-493 Provide a reference for tobacco infiltration.

L504-505 Provide a reference for candidate amplification or provide brief amplification conditions.

L511-512 Confirm that the Phytozome reference corresponds to the R108 line or justify using a different reference line.

L523 I would replace the final "Medicago" with "alfalfa" as used elsewhere in the manuscript.

L528-529 Mention in the legend or Table S6 what the PK acronym means.

Comments on the Quality of English Language

Minor grammatical errors should be checked with careful proof reading. 

Reviewer 2 Report

Comments and Suggestions for Authors

Xing et al. identified MtbZIP60, a novel bZIP transcription factor in Medicago truncatula, as a negative regulator of leaf senescence through transcriptome analysis. The gene was shown to interact with MtWRKY40, playing a crucial role in regulating leaf aging processes, which in turn affects biomass yield and quality in forage crops. These findings present potential targets for the genetic enhancement of alfalfa and other legume forages by modulating leaf senescence. 

Below are my observations and recommending areas for improvement:

1- The abstract lacks sufficient specificity regarding the methods and the statistical robustness of the findings. A more straightforward explanation of how this discovery could directly impact forage crop yield and quality is recommended.

2- The introduction contains excessive detail, with some redundancy regarding the roles of various transcription factor families. Additionally, there is limited discussion of previous studies on Medicago truncatula, which would help contextualize the current research within the broader field. The background could be condensed, emphasizing the specific knowledge gaps addressed by this study. The relevance of Medicago truncatula as a model organism for forage crops and references to prior work in this area should be more explicitly discussed. 

3- Several figures are challenging to interpret due to inadequate explanations of the legends and a lack of clarity regarding the statistical methods employed. It is recommended that figure legends be improved to ensure readers can easily understand the data and that statistical significance annotations be included for all relevant comparisons.

4- There is insufficient detail regarding key methodological steps, such as how the Gene Ontology (GO) and KEGG pathway analyses were conducted. The bioinformatics pipeline, including the specific tools, versions used, and criteria for data quality control, should be clearly outlined.

5- The limitations of the study should be acknowledged more explicitly, particularly the need for validation under field conditions. Future directions should include functional testing of MtbZIP60 in forage crops to assess its potential for real-world agricultural applications.

Comments on the Quality of English Language

The quality of English is fine, but minor editing is required.

Reviewer 3 Report

Comments and Suggestions for Authors

In the current manuscript authors claim that a novel MtbZIP60 gene that plays a negative regulator role in Medicago truncatula leaf senescence was identified after transcriptome analysis and characterized by subcellular localization, cloning and protein-protein interaction assays. This study sounds interesting and may gather relevant biological contributions to the subject. However, I have major concerns about methodology reproducibility. I mean, it lacks almost complete detailing of how essays were carried out and analyzed.

Material and Methods

4.3. RNA Isolation and Gene Expression Analysis

How total RNA quality and quantity were analyzed? Regarding qPCR essays it lacks a description of reaction composition and final volume. Was it performed using technical or biological replicates? Why was the RT-qPCR test performed? It is unclear. How many genes and why were they selected for qPCR test? How were the primers designed?

4.4. Library Construction and Transcriptome Sequencing

There is almost total lack of information about cDNA libraries construction and sequencing, and mostly how the transcriptome was analyzed. 

Methodologies from 4.5 to 4.9 are too superficial and completely unreproducible. All these analyses must be sufficiently detailed so that other researchers may reproduce them.

Results

Authors state that the expression of MtORE1 and MtCAB1 genes was analyzed. I could not find any description in the methodology. 

Libraries length, QC values and mapping percentage were not reported. This must be included at least as supplementary material. 

Round 2

Reviewer 3 Report

Comments and Suggestions for Authors

All my concerns were sufficiently met.